



# Leaf wax *n*-alkane pattern and compound-specific $\delta^{13}$C of plants and topsoils from semi-arid Mongolia

Julian Struck[1], Marcel Bliedtner[1,2], Paul Strobel[1], Jens Schumacher[3], Enkhtuya Bazarradnaa[4], and Roland Zech[1]

[1]Institute of Geography, Friedrich-Schiller-University Jena, Löbdergraben 32, 07743 Jena, Germany
[2]Institute of Geography and Oeschger Centre for Climate Change Research, University of Bern, Hallerstrasse 12, 3012 Bern, Switzerland
[3]Institute of Mathematics, Friedrich-Schiller-University Jena, Ernst-Abbe-Platz 2, 07743 Jena, Germany
[4]Institute of Plant and Agricultural Sciences, Mongolian University of Life Sciences, Mongolia

**Correspondence:** Julian Struck (julian.struck@uni-jena.de)

**Abstract.** *n*-Alkane pattern and their compound-specific $\delta^{13}$C signatures are valuable proxies for paleoenvironmental reconstructions. So far, their potential has not been investigated in semi-arid to arid regions. We have therefore analysed the leaf wax *n*-alkanes and their compound-specific $\delta^{13}$C of five plant species (*Poaceae, Cyperaceae, Artemisia spp., Caragana spp.* and *Larix sp.*), and topsoils $(0 - 5\,\text{cm})$ along two transects in central and southern Mongolia.

Grasses depict a distinct dominance of the *n*-$C_{31}$ homologue, whereas Caragana spp. and Artemisia spp. are dominated by *n*-$C_{29}$. Larix sp. is characterized by the mid-chain *n*-alkanes *n*-$C_{23}$ and *n*-$C_{25}$. From plant to soil, *n*-alkane homologue pattern show the potential to differentiate between grass covered sites from those covered by Caragana spp. *n*-Alkane concentrations and OEP values of the topsoils are distinctly influenced by mean annual temperature, mean annual precipitation and aridity, likely reflecting the degree of *n*-alkane degradation and biomass production. In contrast, the *n*-alkane average chain-length

and the *n*-alkane ratio ($n$-$C_{31}/n$-$C_{29} + n$-$C_{31}$) are not affected by climatic parameters. The compound-specific $\delta^{13}$C signatures are strongly corelated to climate, showing a significant enrichment with increasing aridity, indicating the effect of water use efficiency. Our calibration results suggest that long-chain *n*-alkanes and their compound-specific $\delta^{13}$C signatures have great potential to reconstruct paleoenvironmental and -climatic conditions when used in sediment archives from Mongolia.

## 1 Introduction

Leaf wax biomarkers such as long-chain *n*-alkanes (*n*-$C_{25}$ - *n*-$C_{35}$) are produced in the plant cuticle as a protection layer against environmental stress and become synthesized by the polyketide biosynthetic pathway resulting in a distinct odd-over-even predominance (OEP) (Eglinton and Hamilton, 1967; Shepherd and Wynne Griffiths, 2006). Due to their water insolubility, chemical inertness and relative resistance against biochemical degradation, leaf wax *n*-alkanes stay well preserved in sediment archives over geological timescales and serve as valuable biomarkers for former environmental conditions (Eglinton and Eglin-

ton, 2008). During the last decades, leaf wax *n*-alkanes have increasingly been used for paleoenvironmental reconstructions in lake sediments (Schwark et al., 2002; Aichner et al., 2017; Rach et al., 2017; Sun et al., 2016), marine sediments (Rommer-





skirchen et al., 2006; Castañeda et al., 2009; Schefuss et al., 2005), loess-paleosol sequences (Schäfer et al., 2018; Zech et al., 2013; Häggi et al., 2019) and fluvial sediment-paleosol sequences (Bliedtner et al., 2018b).

The relative homologue distribution of leaf wax $n$-alkanes has been used as a chemotaxonomic marker to differentiate among

vegetation forms and thus reconstruct paleovegetation: the $n$-alkanes $n$-$C_{27}$ and $n$-$C_{29}$ are thought to be mainly produced by deciduous trees/shrubs, whereas $n$-$C_{31}$ and $n$-$C_{33}$ are mainly produced by grasses/herbs (Bliedtner et al., 2018a; Schäfer et al., 2016; Vogts et al., 2009). The compound-specific $\delta^{13}$C isotopes of leaf wax $n$-alkanes have also been used for reconstructing changes in the vegetation composition of $C_3$ ($-20$ to $-35‰$) and $C_4$ ($-10$ to $-14‰$) plants (Castañeda et al., 2009; Lane, 2017; Rao et al., 2016; Rommerskirchen et al., 2006), and give additional paleoclimatic information for $C_3$ plants (Aichner

et al., 2010b, a; Schäfer et al., 2018). Several studies have shown a strong correlation between the $\delta^{13}$C leaf wax signal of $C_3$ plants and water use efficiency (WUE) that is influenced by precipitation, temperature and evapotranspiration and describes the stomata conductance of a plant to avoid water loss (Diefendorf and Freimuth, 2017; Farquhar et al., 1982; Rao et al., 2017). Thus, warmer/dryer conditions cause an increase of WUE resulting in an enrichment of leaf wax $\delta^{13}$C and cooler/wetter conditions *vice versa* (Aichner et al., 2015; Castañeda et al., 2009; Diefendorf and Freimuth, 2017). Although leaf wax $n$-

alkane homologue distributions and compound-specific $\delta^{13}$C isotopes have been increasingly used in sediment archives for paleoenvironmental reconstructions within the last years, they need to be calibrated regionally on recent reference material before any paleoenvironmental reconstruction can be made.

The need for regional calibrations has been emphasized by the fact that Bush and McInerney (2013) questioned whether leaf wax $n$-alkane homologue pattern can discriminate between modern vegetation forms on a global scale while several regional

studies found them discriminating on a regional scale. Although the most abundant homologues differ from region to region, a good discrimination power has been reported from Europe (Schäfer et al., 2016; Zech et al., 2009, 2010), the Caucasus region (Bliedtner et al., 2018a), North- and South America (Diefendorf et al., 2015; Feakins et al., 2016; Lane, 2017) and the African rain forest and savanna (Vogts et al., 2009).

The compound-specific $\delta^{13}$C signature of leaf wax $n$-alkanes are strongly correlated to temperature, as been shown along a

400 mm isohyet in China, with strongest correlations observed for the average temperature of June, July and August (Wang et al., 2018b). Wang et al. (2018b) further indicate a strong linear relationship between the $\delta^{13}$C signature of $n$-$C_{29}$ and $n$-$C_{31}$, identifying both homologues as product of the same biosynthetic process.

However, when interpreting leaf wax $n$-alkanes and their compound-specific $\delta^{13}$C isotopes several potential pitfalls such as species-specific and intra-leaf variations (Diefendorf et al., 2011; Gao et al., 2015), the influence of environmental and climatic

factors (Diefendorf and Freimuth, 2017; Farquhar et al., 1982; Hoffmann et al., 2013; Rao et al., 2017; Tipple et al., 2013; Carr et al., 2014), and $n$-alkane degradation (Buggle et al., 2010; Brittingham et al., 2017; Li et al., 2018a) have to be considered and accounted for. So far, such regional calibration studies on recent leaf wax homologue pattern and compound-specific $\delta^{13}$C isotopes and potential biases of the vegetational/climatic leaf wax signal are scarce for semi-arid regions and do not exist for Mongolia.

This study investigates leaf wax $n$-alkane homologue pattern and compound-specific $\delta^{13}$C isotopes of modern plants and topsoils from semi-arid Mongolia to evaluate their potential for regional paleovegetation and -climate reconstructions. More





specifically, we tested the chemotaxonomic potential of leaf wax homologue pattern from five dominant plant species and whether their homologue distribution can be used to discriminate between woody shrubs and grasses/herbs on a regional scale. Moreover, we investigate differences in the compound-specific $\delta^{13}$C signature of leaf waxes from plants and topsoils, and how

the plant signal is incorporated into the topsoil. Since only the topsoils represent an averaged decadal leaf wax signal, we check for potential climatic influences by correlating the leaf wax homologue pattern and $\delta^{13}$C with mean annual temperature (MAT), mean annual precipitation (MAP) and the aridity index (AI). While we test that the homologue pattern are not biased by climatic influences, we test the potential of leaf wax $\delta^{13}$C to reflect on MAT, MAP and aridity, and thus climate. Therefore, our calibration results will be a base for future robust paleovegetational and -climate reconstructions in semi-arid Mongolia

using leaf wax *n*-alkanes from regional sediment archives. Such lacustrine, loess-paleosol and colluvial sediment archives were increasingly investigated in semi-arid Mongolia and could potentially be used for future leaf wax based paleoenvironmental reconstructions (Klinge et al., 2017; Rudaya and Li, 2013; Peck et al., 2002; Wang et al., 2011; Prokopenko et al., 2007).

## 2    Material and methods

### 2.1    Geographical setting and sampling

Semi-arid Mongolia is a highly continental region characterized by harsh/long winters and hot/short summers (Dashkhuu et al., 2015). Mongolia is located at the interface of three major atmospheric circulation systems controlling the regional climate (Fig. 1). The summer climate is dominated by the East Asian Summer Monsoon (EASM) and the Westerlies that provide most of the yearly moisture and precipitation during the summer month, i.e. 75% of the annual precipitation falls in June, July and August (Wang and Feng, 2013; Rao et al., 2015). The dry and cold winter climate is dominated by the Siberian high that mostly

blocks the moisture supply from the Westerlies during winter (Peck et al., 2002). The Mongolian climate has a north-south gradient in MAT and MAP, with increasing MAT from north to south and MAP *vice versa* (Harris et al., 2014, Fig. 2A, B). This north-south gradient in temperature and precipitation is further reflected by the AI (Fig. 2C) and in the distribution of regional vegetation biomes with taiga and mountain/forest steppe in Northern and Central Mongolia and steppe/desert steppe in Southern Mongolia (Hilbig, 1995; Klinge and Sauer, 2019).

For this study, we sampled topsoils $(0-5\,\text{cm})$ along a north-south transect (transect I, see Fig. 1 for location) in June 2016 as well as plants and topsoils $(0-5\,\text{cm})$ along an east-west transect (transect II, see Fig. 1 for location) in July/August 2017. Along transect II, the topsoils were sampled together with the dominant plant species, which comprise the woody shrub *Caragana spp.*, the grasses/herbs *Poaceae, Cyperaceae* and *Artemisia spp.*, and the coniferous tree *Larix sp.*. Transect I covers a MAT and MAP range from $-1.7$ to $5.5\,°\text{C}$ and 99 to $276\,\text{mm/a}$, whereas Transect II covers a range from $-7.3$ to $-0.5\,°\text{C}$ and 210.8

to $276.2\,\text{mm/a}$ (Fig. 2; Fick and Hijmans, 2017).





## 2.2 Leaf wax analysis

### 2.2.1 Leaf wax extraction and chromatography

Total lipids of the topsoils ($\sim 35\,\mathrm{g}$) from transect I were extracted at the University of Bern, Switzerland, using accelerating solvent extraction (Dionex ASE 200: $6.9\,\mathrm{MPa}, 100\,^{\circ}\mathrm{C}$) with $60\,\mathrm{ml}$ dichlormethane (DCM):methanol (MeOH) (9:1, v/v) over three extraction cycles as described by Schäfer et al. (2016). Total lipids of topsoils ($\sim 10\,\mathrm{g}$) and plants ($\sim 1\,\mathrm{g}$) from transect II were extracted at the Friedrich-Schiller-University of Jena, Germany, using ultrasonic extraction with $20\,\mathrm{ml}$ DCM:MeOH (9:1, v/v) over three cycles as described by Bliedtner et al. (2018a).

The total lipid extract from both transects was separated over aminopropyl pipette columns (Supelco, $45\,\mu\mathrm{m}$) into (i) an apolar fraction including the $n$-alkanes, (ii) a more polar fraction and (iii) an acid fraction. Subsequently, $n$-alkanes were eluted with $\sim 4\,\mathrm{ml}$ hexane and additionally cleaned over coupled silver-nitrate ($\mathrm{AgNO_3}$) – zeolite pipette columns. $n$-Alkanes were subsequently dissolved in hydrofluoric acid and liquid–liquid recovered with $n$-hexane. Identification and quantification of the $n$-alkanes was performed on an Agilent 7890B gas chromatograph equipped with an Agilent HP5MS column ($30\,\mathrm{m} \times 320\,\mu\mathrm{m} \times 0.25\,\mu\mathrm{m}$ film thickness) and a flame ionization detector (GC-FID). For identification and quantification, external $n$-alkane standards ($n$-alkane mix $n\text{-}C_{21} - n\text{-}C_{40}$, Supelco) were run with each sequence.

### 2.2.2 Compound-specific $\delta^{13}$C analysis

Compound-specific carbon isotopes were measured for the most abundant $n$-alkanes $n$-C$_{29}$ and $n$-C$_{31}$. Isotope measurements were performed on an Isoprime Vision isotope ratio mass spectrometer coupled to a gas chromatograph (Agilant 7890B GC) equipped with an Agilent HP5GC column ($30\,\mathrm{m} \times 320\,\mu\mathrm{m} \times 0.25\,\mu\mathrm{m}$ film thickness) via a GC5 pyrolysis/combustion interface. The GC5 was operating in combustion mode with a CuO reactor at $850\,^{\circ}\mathrm{C}$. Samples were injected in splitless mode and measured in triplicates. $n$-Alkane standards ($n$-C$_{27}$, $n$-C$_{29}$ and $n$-C$_{33}$) with known isotopic composition (Schimmelmann standard, Indiana) were measured as duplicates after every third triplicate. The average standard deviation for the triplicate measurements were $0.1\%o$ the standard deviation for the standards was better than $0.2\%o$ ($n = 102$). Carbon isotopic composition is given in the delta notation ($\delta^{13}$C) versus the Vienna Pee Dee Belemnite standard (VPDB).

### 2.2.3 Data analysis

$n$-Alkane concentrations ($\sum n$-Alkane) are given in $\mu\mathrm{g}\,\mathrm{g}^{-1}$ sediment dry weight and were calculated as the sum of $n$-C$_{25}$ to $n$-C$_{35}$. The OEP was calculated according to Hoefs et al. (2002) and serves as a proxy for degradation with values below five indicating enhanced $n$-alkane degradation (Zech et al., 2009, 2010; Buggle et al., 2010).

$$\mathrm{OEP} = \frac{n\text{-}C_{27} + n\text{-}C_{29} + n\text{-}C_{31} + n\text{-}C_{33}}{n\text{-}C_{26} + n\text{-}C_{28} + n\text{-}C_{30} + n\text{-}C_{32}} \tag{1}$$





The average chain length (ACL) was determined after Poynter et al. (1989) and is commonly used to distinguish between
leaf waxes predominantly produced by deciduous trees and shrubs ($n$-C$_{27}$ and $n$-C$_{29}$) and grasses and herbs ($n$-C$_{31}$ and $n$-C$_{33}$)
(Bliedtner et al., 2018a).

$$\text{ACL} = \frac{27 \cdot n\text{-}C_{27} + 29 \cdot n\text{-}C_{29} + 31 \cdot n\text{-}C_{31} + 33 \cdot n\text{-}C_{33}}{v_{27} + n\text{-}C_{29} + n\text{-}C_{31} + n\text{-}C_{33}} \tag{2}$$

A normalized $n$-alkane ratio was calculated for the most abundant $n$-alkanes $n$-C$_{29}$ and $n$-C$_{31}$.

$$n\text{-Alkane Ratio} = \frac{n\text{-}C_{31}}{(n\text{-}C_{29} + n\text{-}C_{31})} \tag{3}$$

### 2.2.4   Statistical analysis

Differences in $n$-alkane concentration, OEP, ACL and compound-specific $\delta^{13}$C among plant species and between topsoils and
plants were analysed using analysis of variance (ANOVA) followed by pairwise comparisons based on Tukey's honestly sig-
nificant difference. Since the relative homologue distribution of the $n$-alkanes is important for the discrimination between plant
species, the $n$-alkane homologue pattern were analysed as compositional data according to Aitchison (2003). Correlations of $n$-
alkane concentration, OEP, ACL and compound-specific $\delta^{13}$C with environmental parameters were tested using weighted linear
or polynomial regression. The environmental parameters MAT and MAP were derived from the WorldClim 2.0 dataset (1970-
2000, 30s resolution Fick and Hijmans, 2017) and the AI from the Global Aridity Index and Potential Evapo-Transpiration
(ET0) Climate Database v2 (1970-2000, 30s resolution Trabucco and Zomer, 2019). Regressions were tested only for the top-
soils because they represent an averaged leaf wax signal over some decades (Angst et al., 2016) and can thus be correlated
with the WorldClim and the Global Aridity Index data that represents averaged climate data over 30 years. In contrast, plants
only reflect an annual signal of the sampling year 2017 and cannot be correlated with the available climatic parameters from
the WorldClim and the Global Aridity Index dataset, and also annual climate data for the sampling year 2017 are not available.
Model selection was based on hierarchical comparison of models with increasing polynomial order using F ratios. Goodness of
fit of the final models was assessed using weighted $R^2$ values. All statistical analyses were done using the statistical software
system R (R Core Team, 2019), and the package compositions (Boogaart, 2013) for compositional data analysis.

## 3   Results

### 3.1   $n$-Alkane pattern in plants and topsoils

Leaf wax $n$-alkanes are present in all analysed plants and topsoils and show a distinct OEP (Fig. 3). For the analysed plants
from transect II, the most abundant $n$-alkane homologues vary among the plant species: *Poaceae* and *Cyperaceae* tend to
be dominated by $n$-C$_{31}$, and *Artemisia spp.* and *Caragana spp.* by $n$-C$_{29}$. *Larix sp.* has its dominance on the mid-chain $n$-
alkane $n$-C$_{25}$. The topsoils from both transects are mostly dominated by $n$-C$_{31}$ (Fig. 3). Figure 4 illustrates differences in



*n*-alkane concentrations, OEP, ACL and the *n*-alkane ratio of the analysed plant species and topsoils, and table 1 show the corresponding level of significance. *n*-Alkane concentrations are significantly higher in plants than topsoils (p = $5.1e^{-07}$). Plants range from 9 to $2508\,\mu g\,g^{-1}$, with *Caragana spp.* having significantly the highest concentrations, and *Larix sp.* having the

lowest concentrations (Fig 4A, Tab 1). *n*-Alkane concentrations in topsoils range from 0.2 to $59\,\mu g\,g^{-1}$ with transect I having much lower concentrations (Fig 4A). Plants show a wide OEP with values ranging between 4 and 39. Highest OEP valuess are observed for *Caragana spp.*, followed by *Poaceae, Cyperaceae, Artemisia spp.* and *Larix sp.*. Topsoils have generally lower OEP values ranging from 1.5 up to 5.5 for transect I and from 4.8 to 19 for transect II, respectively (Fig 4B). ACL values of plants range from 28.3 to 30.8 with *Larix sp.* showing significantly lower ACLs compared to the other plants (Tab 1). ACL

values for topsoils are higher but in the same range as most plant species, ranging from 29.6 to 30.4 along transect I and between 29.2 and 31.8 along transect II (Fig 4C). The *n*-alkane ratio show values between 0.16 and 0.8. *Poaceae* and *Cyperaceae* tend to have the highest values, *Caragana spp.* and *Larix sp.* the lowest. The topsoils range from 0.42 to 0.75 for both transects (Fig 4D).

## 3.2 Compound-specific $\delta^{13}$C

Compound-specific $\delta^{13}$C isotopes were measured for the most abundant *n*-alkanes *n*-$C_{29}$ and *n*-$C_{31}$ (Fig 5). Plants show consistent values between $-36\permil$ to $-29.5\permil$ for $\delta^{13}C_{29}$, and between $-35.8\permil$ and $-30.3\permil$ for $\delta^{13}C_{31}$, with *Larix sp.* having the most enriched values among all plants. In comparison, topsoils tend to be more enriched and scattered, ranging from $-33.8\permil$ to $-25.6\permil$ for $\delta^{13}C_{29}$, and from $-34.3\permil$ to $-25.2\permil$ for $\delta^{13}C_{31}$ with a distinct enrichment in $\delta^{13}$C along transect I (Fig 5). Compound-specific $\delta^{13}$C isotopes of *n*-$C_{29}$ and *n*-$C_{31}$ differ significantly between topsoils from transect I and transect II but

not significantly between plant species and between plant species and topsoils from transect II (Tab 1).

## 4 Discussion

### 4.1 *n*-Alkane pattern in plants

The plant species from Mongolia show distinct differences in their relative *n*-alkane homologue pattern (Fig. 3). The grasses *Poaceae* and *Cyperaceae* are dominated by *n*-$C_{31}$, whereas the woody shrub *Caragana spp.* is dominated by *n*-$C_{29}$. Those

findings are in line with previous regional studies, which report *n*-$C_{31}$ being mainly produced by grasses/herbs and *n*-$C_{29}$ by deciduous trees and shrubs (Cheung et al., 2015; Wang et al., 2018c; Liu et al., 2018; Bliedtner et al., 2018a). Although *Artemisia spp.* could be expected to be herbaceous with a dominance in *n*-$C_{31}$, our results show a distribution maximum at *n*-$C_{29}$, i.e. it is more similar to the woody shrub *Caragana spp.* than to the grass species. This does not necessarily be contradictory and corroborates the findings of Wang et al. (2018b) from China who reports that *Artemisia* can grow both as herbaceous

and as a woody shrub (e.g. *Artemisia frigida*). In contrast to the other plant species, the coniferous tree *Larix sp.* is dominated by the mid-chain *n*-alkanes *n*-$C_{23}$ and *n*-$C_{25}$ resulting in significant lower ACLs (Fig. 4C, Tab. 1). However, statistically significant differences between the ACL of the other plant species are not evident, although the relative homologue pattern reveal





differences among them (Tab. 1, Fig. 3 and 4C). The ACL of the grasses/herbs *Poaceae* and *Cyperaceae* and *Artemisia spp.* and the woody shrub *Caragana spp.* have only slight differences and a small range between 29.6 and 29.9. This due to a strong ACL

scattering of the grasses/herbs that overlap the ACL of the woody shrubs (Fig. 4C). Thus, a clear chemotaxonomic discrimination between grasses/herbs and woody shrubs is not given by the ACL for the investigated modern plants from Mongolia. A better chemotaxonomic discrimination is provided by the *n*-alkane ratio *n*-$C_{31}$/($n$-$C_{29}$ + $n$-$C_{31}$) that is based on the most abundant homologues *n*-$C_{29}$ or *n*-$C_{31}$. The *n*-alkane ratio significantly separates the grasses *Poaceae* and *Cyperaceae* from the woody shrub *Caragana spp.* and the coniferous tree *Larix sp.* (Fig. 4D). The *n*-alkane ratio of *Artemisia spp.* lies in-between the

woody shrubs and grasses, most likely because of their ability to grow as both herbaceous and woody shrubs (Fig. 4D). This is further expressed statistically as the *n*-alkane ratio of *Artemisia spp.* is equal to those calculated from both *Caragana spp.* and the grass species (Tab. 1). Beside *Artemisia*, the *n*-alkane ratio has the chemotaxonomic potential to discriminate significantly between *Larix sp.* and grasses as well as between *Caragana spp.* and grasses. However, the *n*-alkane homologue pattern from *Larix sp.* with their mid-chain dominance has to be interpreted with caution when comparing species-specific differences to

the long-chain dominated plant species, that is mostly due to the fact that *Larix* only produce small amounts of *n*-alkanes. When incorporated into the soil, the coniferous *n*-alkane signal from *Larix* should become overproportional overprinted by the undergrowth of the grasses/herbs (Schäfer et al., 2016; Diefendorf et al., 2011).

## 4.2 Compound-specific $\delta^{13}$C of plants

The compound-specific $\delta^{13}$C values of the *n*-alkanes from our investigated plants from transect II show consistent $\delta^{13}$C values

among the plant species, except *Larix sp.*, and are in a typical range of $C_3$ plants (Tipple and Pagani, 2007). Although Pyankov et al. (2000) have reported $C_4$ plants in Mongolia among 16 plant families including *Poaceae*, those are not evident along our sampled plant transect. While some $C_4$ plants have been found in the Khangai Mountains, their distribution is mainly limited to the semiarid-steppe and semi-desert areas in southern Mongolia and the Gobi desert, i.e. beyond our plant sampling sites (Pyankov et al., 2000; Su et al., 2011). Statistically significant differences did not exist between the most abundant homologues

*n*-$C_{29}$ and *n*-$C_{31}$ ($p = 1$), indicating that no different fractionation has occured during biosynthesis (Wang et al., 2018b). Consistent $\delta^{13}$C values between the most abundant homologues are in good agreement with compound-specific $\delta^{13}$C analyses on three *Artemisia* species (*Artemisia argyi, capilares* and *scoparia*) along a 400 mm isohyet in China (Wang et al., 2018b). While no significant differences are found between the $\delta^{13}$C values of the grasses/herbs and woody shrubs, only *Larix sp.* is enriched up to 2‰ but reveals no statistical significance (Fig. 5, Tab. 1). Such an enrichment of coniferous trees compared to

other plants might be explained by differences in species-specific fractionation (Diefendorf et al., 2015).

## 4.3 Comparing *n*-alkane pattern and compound-specific $\delta^{13}$C of plants versus topsoils

### The leaf wax signal from plants to topsoils along transect II

Along transect II, modern plants have higher *n*-alkane concentration than the topsoils, with *Artemisia spp.* and *Caragana spp.* having significantly higher *n*-alkane concentrations than the respective topsoils (Fig. 4A, Tab. 1). Thus, the lower *n*-alkane





concentration in the topsoils indicate that *n*-alkanes become diluted during the incorporation from plant biomass into the
soil (Fig. 4A). Likewise, the odd-over-even predominance decreases from plants to soil and indicate enhanced organic matter
degradation (Schäfer et al., 2016; Buggle et al., 2010) and microbial alteration (Schulz et al., 2012) (Fig. 4B). Despite possible
degradation effects during soil developement, the topsoils show distinct OEP values between 4.8 and 19, still indicating a
good preservation (Zech et al., 2009). One exception along transect II in terms of higher *n*-alkane concentration and OEP
is TSC10 Ah1, showing $59\,\mu g\,g^{-1}$ and 19, respectively (Fig. 1, sampling site 25). Site TSC10 is characterized by stagnating
soil conditions with a distinct organic rich topsoil, limiting organic matter degradation and microbial alteration of *n*-alkanes
(Hoefs et al., 2002). Thus, TSC10 Ah1 remains exceptional and not comparable to the other topsoils from transects II. Overall,
decreasing concentrations and OEP values from plants to topsoils are in good agreement with other regional studies (Bliedtner
et al., 2018a; Howard et al., 2018; Li et al., 2018b; Schäfer et al., 2016; Zech et al., 2009). For the topsoils, *n*-$C_{31}$ is on average
the most abundant *n*-alkane homologue, indicating a typical *n*-alkane pattern produced by grasses (Bliedtner et al., 2018a,
Fig. 3). The only exceptions are sites covered with *Caragana spp.* ($n = 8$) where higher amounts of *n*-$C_{29}$ are evident within
the respective topsoils and the two *Caragana* covered topsoils TLC4 Ah1 and TLC6 Ah1 (Fig. 1, sampling sites 40 and 42)
even show a dominance of *n*-$C_{29}$ (Suppl. Mat.). Thus, the dominant *n*-$C_{29}$ signal produced by the woody shrubs is also reflected
in the respective topsoil. This is further expressed by lower ACLs and *n*-alkane ratios for those topsoils, which explains the
scattering towards *n*-$C_{29}$ in ACL and 0.4 for the *n*-alkane ratio, respectively (Fig. 4C, D). At sites with *Larix*, the mid-chain
length dominance of *Larix sp.* is not reflected in the respective topsoils, which are mainly dominated by *n*-$C_{31}$ *n*-alkanes. Thus,
*n*-alkanes from *Larix* must become strongly diluted from plant to topsoil and the topsoils reflect mostly the *n*-alkanes from
the grassy undergrowth (Schäfer et al., 2016). Compared to the plants, compound-specific $\delta^{13}$C isotopes of the topsoils are
slightly more enriched (Fig. 5, Tab. 1), which is in line with previous studies and might reflect an enrichment by diagenesis
from litter to topsoil or a change in vegetation composition (Wu et al., 2019, and references therein). Environmental information
of the plants compound-specific $\delta^{13}$C signal only reflects one vegetation period, whereas the topsoils compound-specific $\delta^{13}$C
signal reflects environmental variability on decadal timescales, which might explain the topsoil $\delta^{13}$C enrichment. However,
one topsoil (TLC4 Ah1) shows strongly enriched $\delta^{13}$C values up to $\sim -25\%_o$. Such an enrichment might be explained by *n*-
alkane contributions from succulent plants, which tend to have more enriched $\delta^{13}$C values within the range of $C_3$ plants (Boom
et al., 2014). Succulents were growing on stone rich, thin topsoils in the catchment of Lake Telmen (Fig. 1). For comparison,
we sampled the succulent *Orostachys malacophylla* from the Telmen catchment and analysed their compound-specific $\delta^{13}$C
isotopes that yield $-24.7\%_o$ for *n*-$C_{29}$ ($n = 1$) and $-25.03\%_o$ for *n*-$C_{31}$ ($n = 1$). Thus, increased inputs of succulent $\delta^{13}$C might
be able to explain the more enriched values in the Telmen catchment and the extreme value of $\sim -25\%_o$ from site TLC4.

**The leaf wax signal along both transects**

The topsoils of both Mongolian transects show distinct differences in concentration and OEP, which are higher along transect
II and decrease along transect I. This is mostly due to the fact that *n*-alkane production and degradation is influenced by the
climatic gradient along transect I (see chapter 4.4 and Fig 2 for more detailed discussion). Beside some *n*-$C_{29}$ dominated sites
with *Caragana*, the ACL and the *n*-alkane ratio show the dominance of *n*-$C_{31}$ *n*-alkanes which indicate their origin from the





grasses *Poaceae* and *Cyperacea* (Fig. 4, Bliedtner et al., 2018a; Schäfer et al., 2016; Vogts et al., 2009; Zech et al., 2010).

This is further expressed by the results of ANOVA, because the ACL and *n*-alkane ratios from topsoils and grasses are not statistically different (Tab. 1).

### 4.4 Climatic influences on topsoil *n*-alkane pattern and compound-specific $\delta^{13}$C

To test potential climatic influences on our *n*-alkane proxies, we correlate them with MAT, MAP and AI (Fig. 6). The *n*-alkane pattern show that *n*-alkane concentrations in topsoils and their preservation (OEP) are correlated to climate parameters. We

detected MAT ($R^2 = 0.584, p = 0.0077$) as the strongest climatic control parameter on *n*-alkane concentrations for the topsoils from both transects. Correlations of *n*-alkane concentration with MAP ($R^2 = 0.372, p = 0.01$) and AI ($R^2 = 0.258, p = 0.0031$) are likewise strong and indicate a significant non-linear correlation (Fig. 6). The negative correlation with MAT and positive correlations with MAP and AI are in agreement with climatic control and the fact that higher temperatures reduce the decarbonylation pathway and the formation of *n*-alkanes Shepherd and Wynne Griffiths (2006). More importantly, lower *n*-

alkane concentrations probably indicate reduced biomass production and enhanced *n*-alkane degradation in topsoils. The latter is well reflected by the OEP and strongly correlated with climatic parameters (Fig. 6). Along transect I (sampling sites $1 - 17$), this degradation impact even intensifies when combined with livestock grazing (Kölbl et al., 2011, and references therein). Previous studies have shown correlations between the production of the most abundant homologues and climate, i.e. common vegetation proxies such as the ACL and *n*-alkane ratio should reflect changes in MAT since plants tend to produce longer

*n*-alkanes as protection against water loss (Sachse et al., 2006; Tipple et al., 2013; Bush and McInerney, 2013; Feakins et al., 2016; Wang et al., 2018b, a). We find only weak correlation of ACL with MAT ($R^2 = 0.169, p = 0.018$) and no correlations with MAP and AI (Fig. 6). The *n*-alkane ratio show weak and non-significant correlations with climate parameters. In contrast, compound-specific $\delta^{13}$C of the topsoils correlate significantly with climatic parameters. Our results show an enrichment in $\delta^{13}$C with increasing temperature, aridity and decreasing precipitation (Fig. 6). This climate induced enrichment in compound-

specific $\delta^{13}$C follows mainly the north-south gradient in decreasing MAP and increasing MAT along transect I from central Mongolia into the Gobi Desert. The only exceptions are the extreme values near Lake Telmen, which are mostly due to the input of the succulent *Orostachys malacophylla* with strongly enriched values. As already proposed by Diefendorf et al. (2010), MAP is a strong predictor on $\delta^{13}$C (*n*-C$_{29}$: $R^2 = 0.683, p = < 1e - 04$, *n*-C$_{31}$: $R^2 = 0.343, p = < 1e - 04$), which is further expressed in strong linear correlations with the AI (Fig. 6). For $\delta^{13}$C$_{29}$ MAT seems to be even stronger than MAP showing a

strong non-linear correlation ($R^2 = 0.691, p = 0.0053$), whereas $\delta^{13}$C$_{31}$ shows a linear but not as strong correlation, than MAP and AI (Fig. 6). Temperature and precipitation are both strong control parameter on water availability and evapo-transpiration, effecting the photorespiration of C$_3$ plants and thus, the WUE (Tipple and Pagani, 2007; Diefendorf and Freimuth, 2017).





# 5 Conclusions

This study investigates leaf wax $n$-alkane homologue pattern and compound-specific $\delta^{13}$C of modern plants and topsoils from
semi-arid Mongolia to test their chemotaxonomic potential and dependency on climate. Our results provide the first regional calibration of leaf wax $n$-alkanes for semi-arid Mongolia with the following results:

**i.** Mongolian plants show distinct differences in their relative $n$-alkane homologue pattern. $n$-Alkanes from the grasses (*Poaceae* and *Cyperaceae*) are clearly dominated by $n$-C$_{31}$, whereas the woody shrub *Caragana spp.* is dominated by $n$-C$_{29}$. Since *Artemisia* species can grow both as herbaceous and woody shrubs, *Artemisia spp.* shows not a typical $n$-C$_{31}$ dominance
but is rather more equal to *Caragana spp.* with a dominance in $n$-C$_{29}$. *Larix sp.* is dominated by the mid-chain $n$-alkanes $n$-C$_{23}$ and $n$-C$_{25}$. Since, *Larix sp.* produces only very few amounts of $n$-alkanes they are not useful for reconstructing vegetation changes in *Larix sp.*. Although the ACL reveal no potential to discriminate between plant species, the most abundant $n$-alkanes $n$-C$_{29}$ and $n$-C$_{31}$ allow to discriminate between woody shrubs and grasses, which is expressed in the $n$-alkane ratio $n$-C$_{31}$/$n$-C$_{29}$ + $n$-C$_{31}$.

**ii.** From plants to soils of transect II, the decrease of $n$-alkane concentrations and OEP values indicate $n$-alkane dilution with mineral soil components and ongoing $n$-alkane degradation. The $n$-alkane pattern of the topsoils are mainly characterized by a dominance of $n$-C$_{31}$, indicating dominant input from grasses. *Caragana* covered sites tend to reflect the homologue pattern of *Caragana spp.*, with $n$-C$_{29}$ being the most dominant $n$-alkane. Topsoils under *Larix sp.* are dominated by the input from the grassy undergrowth. There are no significant differences in compound-specific $\delta^{13}$C between plant species
and topsoils. Topsoils tend to be 2‰ enriched compared to the plants, indicating diagenesis from litter to topsoil and the climatic influence.

**iii.** $n$-Alkane concentrations and OEP values from Mongolian topsoils are significantly correlated to climatic parameters and decreasing with increasing MAT and decreasing MAP. In contrast, our data indicate that the $n$-alkane homologue pattern from the topsoils are not influenced by climatic parameters and thus, the $n$-alkane ratio can reliably be used to detect and
reconstruct differences between the vegetation forms of grasses and woody shrubs. For compound-specific $\delta^{13}$C of the topsoils, strong correlations exist with increasing MAT and decreasing MAP, indicating an enhanced enrichment in $\delta^{13}$C with increasing aridity and drought stress. Thus it can be a valuable proxy for to reflect on climate variations.

Our results show, that the $n$-alkane homologues $n$-C$_{29}$ and $n$-C$_{31}$ have the chemotaxonomic power to differentiate between grasses and the woody shrub *Caragana spp.*. Future studies on plant $n$-alkane homologues should include a detailed iden-
tification of plant species, to reveal the full power of the $n$-alkane ratio as a vegetation proxy. This is particularly the case for different *Artemisia* species, which can so far not be separated from grasses and woody shrubs. While the homologue patterns are not biased by climatic influences, compound-specific $\delta^{13}$C indicate a strong climatic dependency. Thus, $n$-alkanes and their compound-specific $\delta^{13}$C signatures can be used as valuable proxies for future leaf wax based paleoenvironmental reconstruction in sediment archives from semi-arid Mongolia.






*Data availability.* The dataset that is used in this study is available in the supplementary material.


*Competing interests.* JS, MB, PS, JSch, EB and RZ declare that they have no conflict of interest.

*Acknowledgements.* We thank our logistic partners in Mongolia and all field trip participants (2017 & 2018). For assistance in the laboratory, we would like to thank M. Wagner and F. Freitag, as well as M. Zech for scientific discussion.





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

**Figure 1.** Map of Mongolia (SRTM DEM). The black circles mark the sampling sites along transect I and II. Black arrows indicate the influence of three major atmospheric circulation systems: the Westerlies, the East Asian Summer Monsoon and the Siberian High. Submaps show (A.) the catchment of Lake Telmen, (B.) an altitude transect near the catchment of Lake Tsagaan and (C.) the catchment of Lake Ugii in more detail.





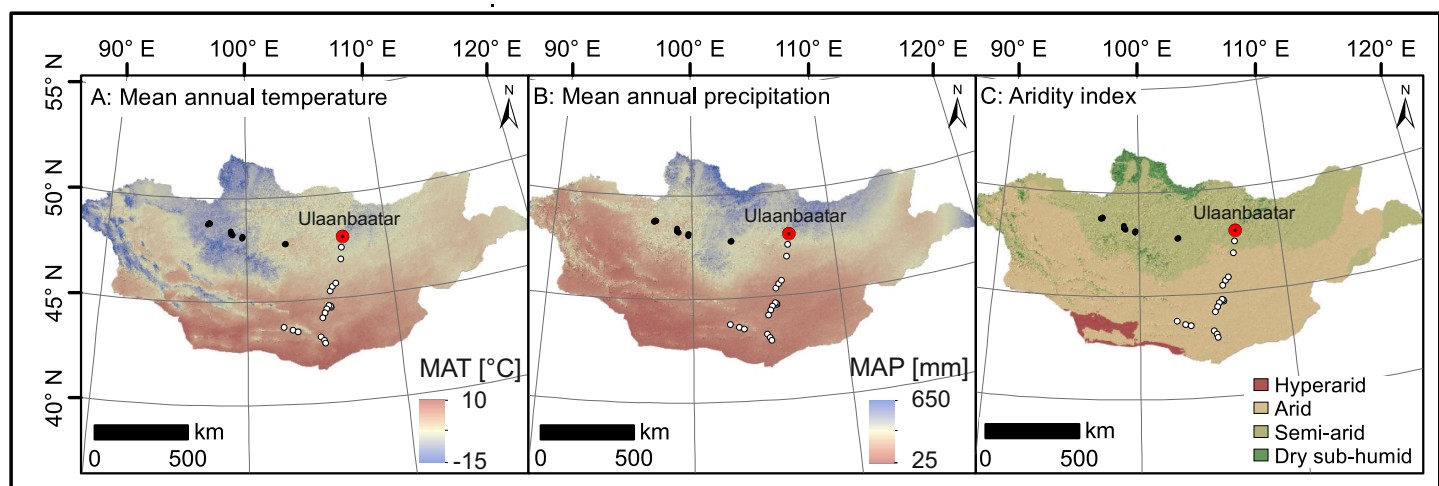

**Figure 2.** Climate and environmental conditions of Mongolia. Mean annual temperature (A), mean annual precipitation (B) and the aridity index (C). MAT and MAP are based on the WorldClim Dataset of Fick and Hijmans (2017), the AI is based on the Global Aridity Index and Evapo-Transpiration Climate Database v2 of Trabucco and Zomer (2019).





**Figure 3.** *n*-Alkane homologues pattern of plants (transect II) and topsoils (transect I and II) from Mongolia. Plants originate from transect II. The bars show the mean values ± standard deviations.





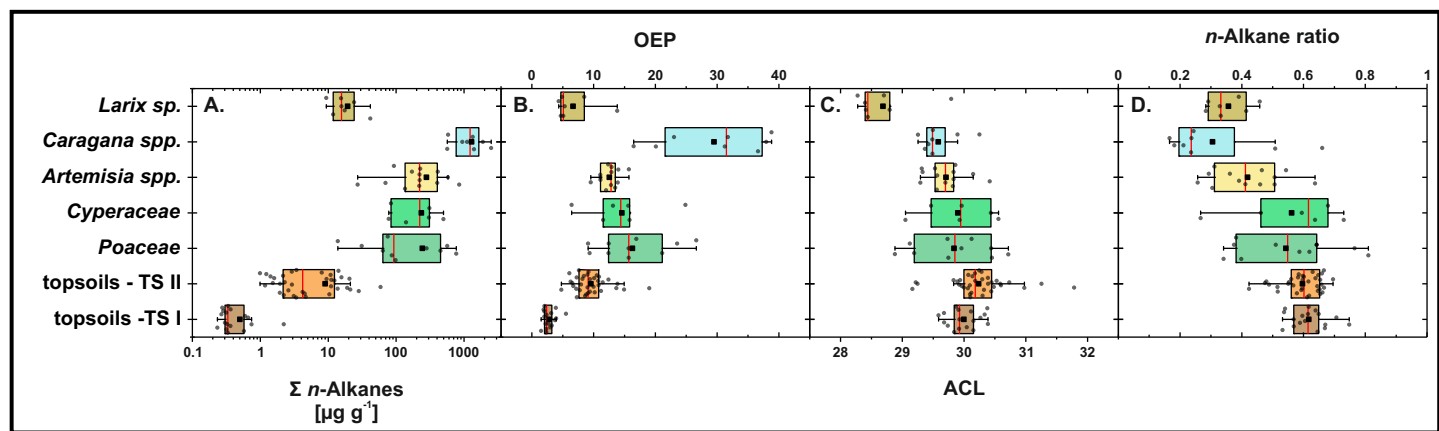

**Figure 4.** *n*-Alkane concentration (A.), OEP (B.), ACL (C.) and *n*-alkane ratio (D.) of plants and topsoils from Mongolia (n = *Larix sp.* = 7, *Cyperaceae* = 6, *Poaceae* = 10, *Caragana spp.* = 8, *Artemisia spp.* = 13, topsoils TS I = 17, topsoils TS II = 35). The boxplots indicate median values (red lines), mean values (black squares), interquartile ranges with lower (25%) and upper (75%) quartiles (box), outlayers (whiskers) and investigated samples (grey circles). See Tab. 1 for statistics.





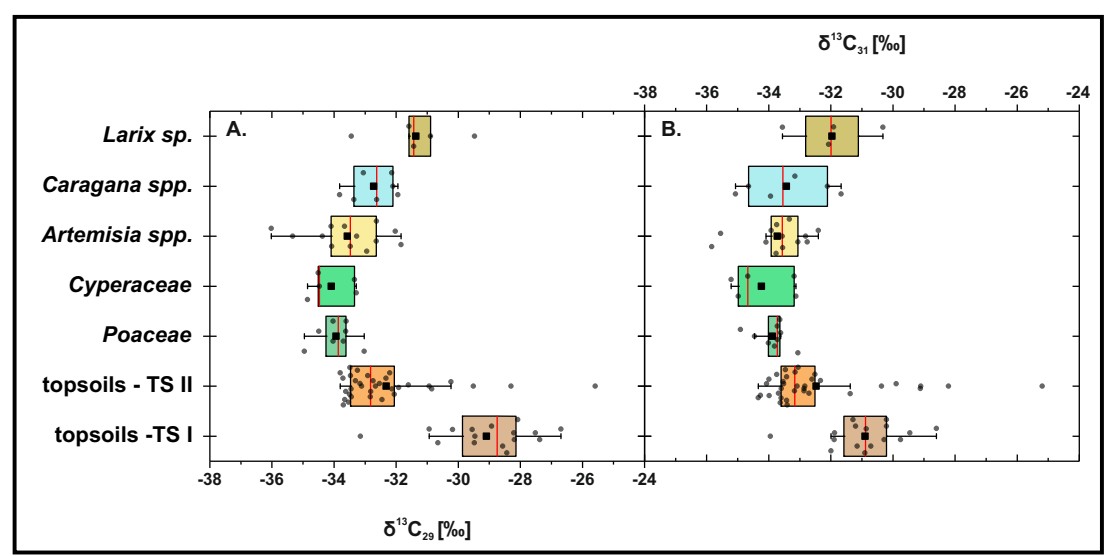

**Figure 5.** Compound-specific $\delta^{13}$C of plants and topsoils from Mongolia. (A.) Compound-specific $\delta^{13}C_{29}$ (n: *Larix sp.* = 5, *Cyperaceae* = 5, *Poaceae* = 8, *Caragana spp.* = 7, *Artemisia spp.* = 13, topsoils TS I = 16, topsoils TS II = 34). (B.) $\delta^{13}C_{31}$ (n: *Larix sp.* = 4, *Cyperaceae* = 5, *Poaceae* = 9, *Caragana spp.* = 6, *Artemisia sp.* = 13, topsoils TS I = 16, topsoils TS II = 34). The boxplots indicate median values (red lines), mean values (black squares), interquartile ranges with lower (25%) and upper (75%) quartiles (box), outlayers (whiskers) and investigated samples (grey circles). See Tab. 1 for statistics.





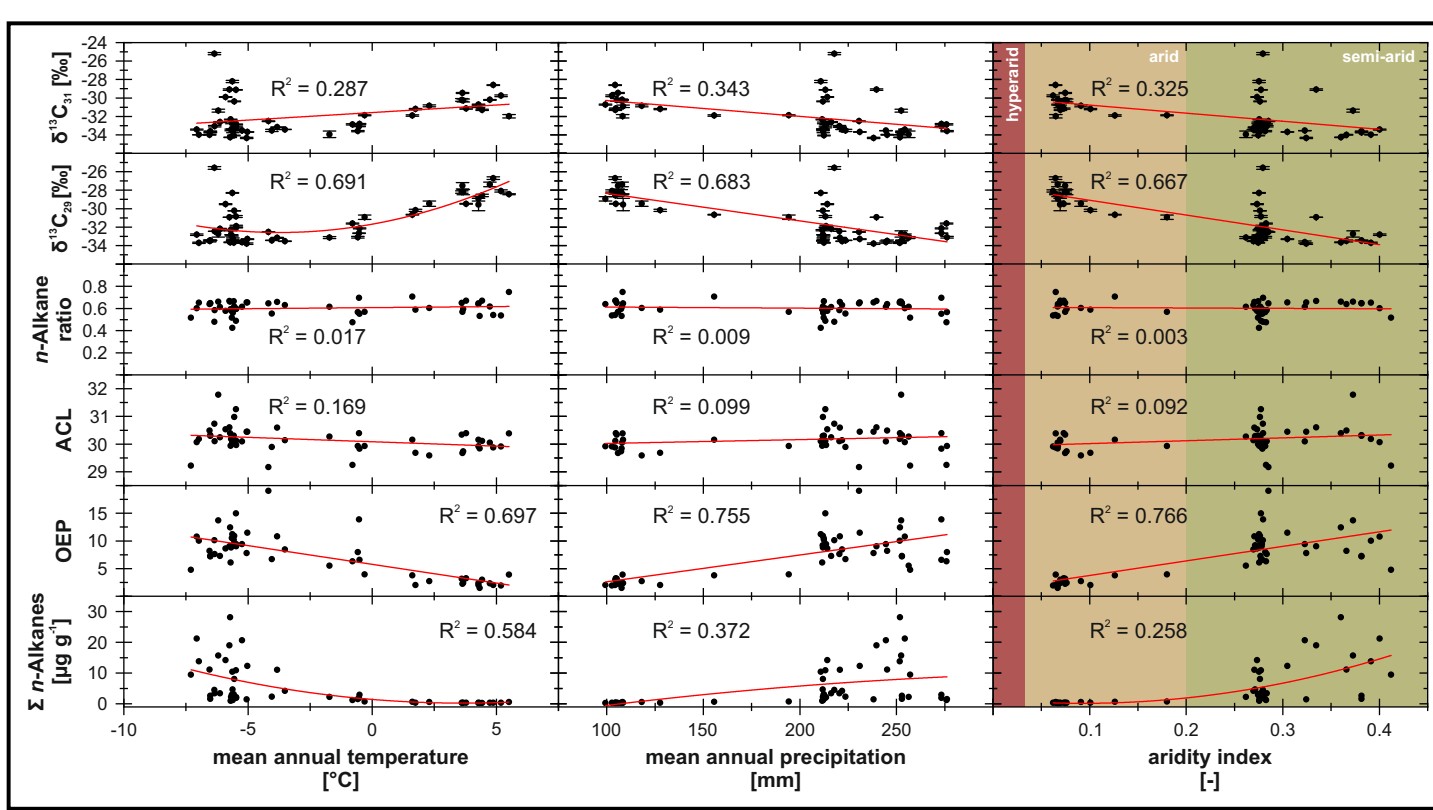

**Figure 6.** *n*-Alkane concentration, OEP, ACL, *n*-alkane ratio and compound-specific $\delta^{13}C$ (*n*-$C_{29}$ and *n*-$C_{31}$) from Mongolian topsoils plotted against climatic parameters (MAP, MAT, AI). Red trend lines illustrate linear or polynomial regressions.





**Table 1.** ANOVA p-values indicating differences among plant species and between topsoils and plants for *n*-alkane pattern and compound-specific $\delta^{13}C$ (*n*-$C_{29}$ and *n*-$C_{31}$). Bold values indicate significance ($\alpha = 0.05$). TS = topsoils.

| | $\sum$*n*-Alkane | OEP | ACL | *n*-Alkane ratio | $\delta^{13}C_{29}$ | $\delta^{13}C_{31}$ |
|---|---|---|---|---|---|---|
| *Poaceae - Cyperaceae* | 1.000 | 0.979 | 1.000 | 0.999 | 1.000 | 0.999 |
| *Larix sp. - Cyperaceae* | 0.505 | **0.032** | **0.000** | **0.010** | 0.203 | 0.315 |
| *Poaceae - Artemisia spp.* | 0.999 | 0.392 | 0.986 | 0.069 | 0.998 | 1.000 |
| *Poaceae - Caragana spp.* | **0.000** | **0.000** | 0.861 | **0.000** | 0.814 | 0.995 |
| *Poaceae - Larix sp.* | 0.325 | **0.001** | **0.000** | **0.007** | 0.164 | 0.381 |
| *Cyperaceae - Artemisia spp.* | 0.999 | 0.948 | 0.959 | 0.079 | 0.995 | 0.991 |
| *Cyperaceae - Caragana spp.* | **0.000** | **0.000** | 0.802 | **0.000** | 0.812 | 0.965 |
| *Caragana spp. - Artemisia spp.* | **0.000** | **0.000** | 0.991 | 0.185 | 0.928 | 0.999 |
| *Larix sp. - Artemisia spp.* | 0.137 | 0.083 | **0.000** | 0.806 | 0.230 | 0.425 |
| *Larix sp. - Caragana spp.* | **0.000** | **0.000** | **0.004** | 0.946 | 1.826 | 0.734 |
| *TS Transect II - Poaceae* | 0.063 | **0.000** | 0.204 | 0.801 | 0.085 | 0.198 |
| *TS Transect II - Cyperaceae* | 0.259 | 0.075 | 0.658 | 0.99 | 0.162 | 0.219 |
| *TS Transect II - Artemisia spp.* | **0.006** | 0.251 | **0.012** | **0.000** | 0.132 | 0.181 |
| *TS Transect II - Caragana spp.* | **0.000** | **0.000** | **0.009** | **0.000** | 0.994 | 0.799 |
| *TS Transect II - Larix sp.* | 1.000 | 0.573 | **0.000** | **0.000** | 0.818 | 0.996 |
| *TS Transect I - TS Transect II* | **0.003** | **0** | 0.082 | 0.273 | **0** | **0.006** |