# Peer review of "Leaf wax *n*-alkane patterns and compound-specific $\delta^{13}$ C of plants and topsoils from semi-arid/arid Mongolia"

_Biogeosciences, 2019_

## Referee Comment (RC1) · Anonymous Referee #1 · 15 Aug 2019

General comments

The study "Leaf wax n-alkane pattern and compound-specific d13C of plants and top-soils from semi-arid Mongolia" by Julian Struck and colleagues presents novel data on content and isotopic signatures of plant waxes along two climate transects in Mongolia. Generally, the study is written in a clear and understandable way. The methods used are standard within the respective research community and the execution from sampling to data analysis seems robust. Also the presentation of data in terms of figures and tables is clear and straightforward understandable.

A major issue with the language is that the clarity could be improved at instances

when comparisons are made. Here, it is often just stated that something is "higher" or "larger", but it is often missing "compared to what". Most of the time this could be traced from the sentence before, but I would suggest to always add this information in the same sentence for clarity. Thus, please check again the whole manuscript for statements where comparisons are made.

Another major issue concerning the science is that (especially in the conclusions section) contradicting statements are made. Here, it is stated that n-alkane homologue patterns are not influenced by climatic parameters, although they are strongly correlated to OEP, which is actually a numerical representation of (changes in) the homologue patterns. Thus, I suggest to carefully review this section, since in the present form it is unclear and contradictory.

Please find some more specific comments and technical corrections below.

Specific comments

L2 There are previous studies, which looked at n-alkanes in (semi-)arid regions. Just as a singular example (there are others): Feakins and Sessions (2010 in Geochimica et Cosmochimica Acta). Thus, I suggest to remove or rephrase this bold statement.

L29 Please specify "additional paleoclimatic information".

L44-47 The paragraph does not fit very well here. Maybe it would be better to incorporate in the section L24-37?

L117 Please check equation 2 again. It is not clear what "v27" stands for.

L168-170 How did Artemisia grow in your study area: herbaceous or woody shrub? Please specify.

L189 Please give reference to the figure where the data is shown.

L190 Why "except Larix"? Please specify.

L198-199 Please rephrase the sentence: "no significance" is redundant and it could be clearer

L243-267 The whole paragraph shows an extensive use of the word "strong correlation". Could you please state (maybe already in the methods section), when you consider a correlation as "strong"? Is there a $R^2$ threshold which you apply? Please specify.

L252 Please elaborate a bit more on the link with livestock grazing, since this is not obvious.

L265 I would remove the statement that the $R^2$ of 0.683 "seems to be even stronger" than the $R^2$ of 0.691. In my opinion they are similar.

L267 Please define WUE at some point (if not already done). I guess it is "water use efficiency".

L288 It sounds contradicting that you state "the n-alkane homologue patterns from the topsoils are not influenced by climatic parameters, and thus the n-alkane ratio can reliably be used to detect and reconstruct differences between the vegetation forms of grasses and woody shrubs". First, in the sentence before you state that n-alkane concentrations and OEP values are significantly correlated to climatic parameters. Second, what do you mean with n-alkane homologue pattern (the sum of n-alkanes concentrations, ACL, OEP or n-alkane ratio)? Please check again, since in the present form the sentence is unclear and contradictory.

L294 Please specify what you mean with "detailed identification of plant species".

L296 Again also here: It is contradictory when you state that "homologue patterns are not biased by climatic influences", although you show correlations of OEP with climate and describe these as "strong" in your discussion. In the end, OEP etc. are just numerical representations of the homologue pattern. Thus, please clarify the contradiction.

L298 Maybe "can be potentially used", to weaken the conclusion a bit.

Figure 3 The sentence "Plants originate from transect II." Is redundant and can be removed.

Figure 6 Please indicate which regressions are linear and which are polynomial. Also I would suggest to add the p-values along with the R2, to clarify the significance of the fit.

Table 1 Check decimal places in last row

Technical corrections

Title "patterns"

L1 "patterns"

L11 "correlated"

L16 "are synthesized"

L21 check order of references and hyphenation

L67 check order of references

L88 "accelerated"

L89 "dichloromethane"

L102 "Agilent"

L110 delete "sediment", since it is either soil or plant material

L142 "Table 1 shows"

L249 Check brackets on reference

L250 Maybe better: "... OEP, which is strongly correlated ..."

L269 "patterns"

L272 "patterns"

L277 Two points at end of sentence

L288 "decrease with increasing"

L288 "patterns"

L292 delete "for"

L294 Two points at end of sentence

---

## Author Comment (AC1) · 5 Sep 2019

**Author responses to the review of Referee #1 of the Biogeosciences manuscript bg-2019-251: 'Leaf wax *n*-alkane patterns and compound-specific δ¹³C of plants and topsoils from semi-arid Mongolia'**

By Julian Struck, Marcel Bliedtner, Paul Strobel, Jens Schumacher, Enkhtuya Bazarradnaa, Roland Zech

We are thankful to referee #1 for the detailed and constructive comments on our manuscript, and we will revise it accordingly.

Please find below our point-to-point response to the review of referee #1. Referee comments are given in *black italic font,* our response to each point in blue regular font. Resulting changes are given in *blue italic.*

**Anonymous Referee #1:**

**General comments**

*The study "Leaf wax n-alkane pattern and compound-specific d13C of plants and topsoils from semi-arid Mongolia" by Julian Struck and colleagues presents novel data on content and isotopic signatures of plant waxes along two climate transects in Mongolia. Generally, the study is written in a clear and understandable way. The methods used are standard within the respective research community and the execution from sampling to data analysis seems robust. Also, the presentation of data in terms of figures and tables is clear and straightforward understandable.*

→ We are very happy about these positive comments on our manuscript.

*A major issue with the language is that the clarity could be improved at instances when comparisons are made. Here, it is often just stated that something is "higher" or "larger", but it is often missing "compared to what". Most of the time this could be traced from the sentence before, but I would suggest to always add this information in the same sentence for clarity. Thus, please check again the whole manuscript for statements where comparisons are made.*

→ We have checked the manuscript and adjusted sentences with comparisons for clarity.

*Another major issue concerning the science is that (especially in the conclusions section) contradicting statements are made. Here, it is stated that n-alkane homologue patterns are not influenced by climatic parameters, although they are strongly correlated to OEP, which is actually a numerical representation of (changes in) the homologue patterns. Thus, I suggest to carefully review this section, since in the present form it is unclear and contradictory.*

→ You are right, and we have to be more specific in the discussion that we have used the term *n*-alkane homologue patterns in terms of typical *n*-alkane distributions for different plants or plant-groups. E.g. typical n-alkane homologue pattern for grasses/ herbs show a dominance of $n\text{-}C_{31}$ and $n\text{-}C_{33}$ and deciduous trees/shrubs $n\text{-}C_{27}$ and $n\text{-}C_{29}$.

**Specific comments**

*1.)      L2 There are previous studies, which looked at n-alkanes in (semi-)arid regions. Just as a singular example (there are others): Feakins and Sessions (2010 in Geochimica et Cosmochimica Acta). Thus, I suggest to remove or rephrase this bold statement.*

→ We have rephrased this statement and refer especially to the semi-arid/arid regions of Mongolia.

*2.)      L29 Please specify "additional paleoclimatic information".*

→ We added 'about drought stress conditions'

*3.)      L44-47 The paragraph does not fit very well here. Maybe it would be better to incorporate in the section L24-37?*

→ We have incorporated the sentence dealing with the temperature correlation in the section L34-37.

*4.)      L117 Please check equation 2 again. It is not clear what "v27" stands for.*

→ We have corrected it to *n*-C27.

*5.)      L168-170 How did Artemisia grow in your study area: herbaceous or woody shrub? Please specify.*

→ Along transect II, Artemisia plants grow both as herbaceous plants and as perennial 'shrubby' plants with a woody base (e.g. Artemisia frigida). Our dataset includes Artemisia frigida samples and herbaceous Artemisia samples. Unfortunately, most samples could only be determined on a plant genus level (Artemisia spp.). We have included a specification in section 2.1 Geographical setting and sampling: *"Artemisia spp. summarizes different herbaceous species and perennial 'shrubby' species with a woody base (e.g. Artemisia frigida)."*

*6.)      L189 Please give reference to the figure where the data is shown.*

→ Done.

*7.)      L190 Why "except Larix"? Please specify.*

→ We have deleted this statement because Larix sp. is also within the typical range of C3 Plants.

*8.)      L198-199 Please rephrase the sentence: "no significance" is redundant and it could be clearer*

→ We have deleted the term significant and changed the sentence to *"While no differences are found between the $\delta^{13}C$ values of the grasses/herbs and woody shrubs, only Larix sp. is enriched up to 2‰, but still in the range of $C_3$ plants."*

*9.)      L243-267 The whole paragraph shows an extensive use of the word "strong correlation". Could you please state (maybe already in the methods section), when you consider a correlation as "strong"? Is there a R2 threshold which you apply? Please specify.*

→ We have rephrased this paragraph in terms of the word "strong correlation". We have not applied any R² threshold. The goodness of fit is based on the weighted R² values and comparisons were done relative to each other under consideration of the p-values.

10.) L252 Please elaborate a bit more on the link with livestock grazing, since this is not obvious.

→ We agree with referee #1 that the former version was misleading. We have changed the structure of the paragraph. The link to livestock grazing is now described after the variations of n-alkane concentrations. Due to the fact that biomass production decreases with increasing aridity, the n-alkane concentration in topsoils become reduced since less organic material is incorporated into the topsoil. *n*-Alkane concentrations in the topsoils become even more reduced with intensified livestock grazing, because less biomass will be incorporated.

11.) L265 I would remove the statement that the R2 of 0.683 "seems to be even stronger" than the R2 of 0.691. In my opinion they are similar.

→ We agree. The statement has been removed.

12.) L267 Please define WUE at some point (if not already done). I guess it is "water use efficiency".

→ It's defined in the introduction as water use efficiency.

13.) L288 It sounds contradicting that you state "the n-alkane homologue patterns from the topsoils are not influenced by climatic parameters, and thus the n-alkane ratio can reliably be used to detect and reconstruct differences between the vegetation forms of grasses and woody shrubs". First, in the sentence before you state that n-alkane concentrations and OEP values are significantly correlated to climatic parameters. Second, what do you mean with n-alkane homologue pattern (the sum of n-alkanes concentrations, ACL, OEP or n-alkane ratio)? Please check again, since in the present form the sentence is unclear and contradictory.

→ We have used the term *n*-alkane homologue patterns in terms of typical *n*-alkane distributions for different plants or plant-groups. E.g. typical n-alkane homologue pattern for grasses/ herbs show a dominance of $n$-$C_{31}$ and $n$-$C_{33}$ and deciduous trees/shrubs $n$-$C_{27}$ and $n$-$C_{29}$. We have specified our statement and defined what we mean with n-alkane homologue patterns in this case.

14.) L294 Please specify what you mean with "detailed identification of plant species".

→ We agree and changed the sentence to "should include a detailed identification of plants regarding different species of each plant genus."

15.) L296 Again also here: It is contradictory when you state that "homologue patterns are not biased by climatic influences", although you show correlations of OEP with climate and describe these as "strong" in your discussion. In the end, OEP etc. are just numerical representations of the homologue pattern. Thus, please clarify the contradiction.

→ We have changes homologue patterns to ACL and *n*-alkane ratio

16.) L298 Maybe "can be potentially used", to weaken the conclusion a bit.

→ We agree

*17.) Figure 3 The sentence "Plants originate from transect II." Is redundant and can be removed.*

→ This is correct, we removed it.

*18.) Figure 6 Please indicate which regressions are linear and which are polynomial. Also I would suggest to add the p-values along with the R2, to clarify the significance of the fit.*

→ We have added the p-values along with the R² values. Linear and non-linear regressions have now different colours and are described in the Fig caption. We have checked all regressions and based on the p-values we have chosen linear or non-linear regressions.

[Figure]

Figure 6. *n*-Alkane concentration, OEP, ACL, *n*-alkane ratio and compound-specific $\delta^{13}C$ (*n*-$C_{29}$ and *n*-$C_{31}$) from Mongolian topsoils plotted against climatic parameters (MAP, MAT, AI). Red trend lines illustrate linear, black lines polynomial regressions. Bold values indicate significance ($\alpha = 0.05$).

*19.) Table 1 Check decimal places in last row*

→ We have changed it to three decimal places (0.000).

**Technical corrections**

*Title "patterns"* → corrected

*L1 "patterns"* → corrected

*L11 "correlated"* → corrected

*L16 "are synthesized"* → corrected

*L21 check order of references and hyphenation* → We checked the order of all references. Now they are all sorted alphabetically / Hyphenation is done by the Copernicus LATEX template

*L67 check order of references* → We checked the order of all references. Now they are all sorted alphabetically

*L88 "accelerated"* → corrected

*L89 "dichloromethane"* → corrected

*L102 "Agilent"* → corrected

*L110 delete "sediment", since it is either soil or plant material* → adjusted

*L142 "Table 1 shows"* → corrected

*L249 Check brackets on reference* → corrected

*L250 Maybe better: ": : : OEP, which is strongly correlated : : :"* → We agree

*L269 "patterns"* → corrected

*L272 "patterns"* → corrected

*L277 Two points at end of sentence* → corrected

*L288 "decrease with increasing"* → corrected

*L288 "patterns"* → corrected

*L292 delete "for"* → done

*L294 Two points at end of sentence* → corrected

**Additional adjustments**

L146 valuess → values

Figure caption of fig 1 → we changed black cycles to black/whithe cycles

---

## Referee Comment (RC2) · Anonymous Referee #2 · 10 Dec 2019

The authors of this manuscript investigated the link between several environmental parameters (mean annual temperature, mean annual precipitation, and aridity index) and molecular and stable carbon isotope compositions of leaf wax n-alkanes extracted from modern higher plants and topsoils along 2 broad transects in Mongolia. The manuscript provides much needed molecular and stable isotope data for that area and will be of interest to biogeochemists, paleoecologists, and paleoclimatologists studying past climate change in the arid zones of central Eurasia. The manuscript fits within the scope of Biogeosciences Discussions and should be published in this journal provided the authors address the following issues:

[Figure]

MAJOR POINT TO ADDRESS

First, lumping topsoil n-alkane data when looking at Transects I and II

Transect II The data shown in Fig. 4 and Fig. 5 for Transect II has a lot of scatter. The transect includes 3 different areas A, B, and C, with area B corresponding to an altitudinal transect. Could this scatter be the result of additional factors controlling the molecular and d13C data along the altitudinal transect in addition to those that play role along the W-E transect (i.e. A, through B (only sites 22, 23, 24, 25) through C)? Could the altitudinal transect sites be plotting separately?

Transect II + Transect I A similar issue could be the reason for a large scatter in the d13C data in Fig. 6 (top 2 sections). There is a lot of scatter at $\sim$ - 6C MAT, $\sim$ 210 mm MAP, and $\sim$ 0.28 AI. Could this be caused by multiple factors (in addition to those plotted along the X-axis) controlling the d13C values of n-alkanes along and within these transects? Can the data be plotted separately to provide a more nuanced assessment?

OTHER MINOR ISSUES

Line 34 (here and similar issues throughout the manuscript) "an enrichment of leaf wax d13C" d13C values are numbers. Values can't be enriched or depleted. Please re-phrase. 13C-enriched leaf wax perhaps?

Lines 82-83 "the topsoils were sampled together with the dominant plant species, which comprise the woody shrub Caragana spp. ..." How the dominance of these species was assessed? Is there any previous study concerning species distribution in the area covered by this project? Or is it a subjective assessment?

Line 90 "Total lipids ... of plants" What part(s) of plant was(were) extracted? Just leaves or was it together with the stem and roots?

Line 107 "were 0.1 per mil the standard deviation" Is there an "and" missing? Also, does it make sense to report d13C values in Suppl. Mat to the second digit after the

decimal point, if the reported std. dev. is 0.1 per mil, i.e. no better than the first digit after the decimal point?

Lines 110-111 "n-alkane concentrations ... were calculated as the sum of n-C25 and n-C35" Why was n-C23 excluded? It is a major n-alkane in Larix sp.

Lines 118-119 "A normalised n-alkane ratio ... n-C29 and n-C31"; lines 177-178 Please explain the significance of this ratio. If this refers to trees/shrubs vs. grasses, why not to include n-C27 and n-C33, respectively?

Line 165 "in line with previous regional studies" Specify what regions were covered by these studies previously? Is it similar to the region covered in this project?

Line 169 "the findings of Wang et al. (2018b) from China" What part of China? It is a big country with multiple climatic and ecological zones.

Line 202-233 Section "The leaf wax signal from plants to topsoils along transect II" This section could be broken down into several paragraphs to make it easier to follow. Also, why not to give a number for this and the next (starting on line 234) subsections? 4.3.1 and 4.3.2 perhaps?

Lines 223-224 "Compared to the plants, ... d13C isotopes of the topsoils are slightly more enriched" What does "d13C isotopes" mean? Please rephrase. Also, I don't see this from the graphs in Fig. 5. The d13C values of n-C29 alkane in soils aren't really that different from those in Caragana and Larix spp. The absence of this "enrichment" is particularly evident when looking at the d13C values of n-C31.

Line 237 "chapter 4.4" I'd call it a section rather than a "chapter".

Lines 248-249 "are in agreement with climatic control and the fact that higher temperatures reduce the decarboxylation pathway and the formation of n-alkanes (Shepherd and Griffiths 2006)" I don't think this explanation works here. The cited paper evaluates the effect of stress on various factors that control leaf wax biosynthesis within plants. The subject matter of this section is n-alkane content of soils. There are multiple other

reasons, in addition to the biochemical ones within the plant, that could play a role in the distribution of n-alkanes along the transects studied.

Lines 257-258 "In contrast, compound-specific d13C correlated significantly with climatic parameters." There is a lot of noise in the d13C data. Please discuss possible reasons for the scatter. See the MAJOR POINT above.

Line 272 "Mongolian plants show" It is a peculiar way of referring to these plants. Using their species names and mentioning that they were sampled in Mongolia would be a better way of describing them.

Lines 276-277 "for reconstructing vegetation changes in Larix sp." Sounds awkward. Please re-phrase.

FIGURES

Figure 1 Please remind the reader what SRTM DEM stands for so that there is no need to look for this information in the text. Also, make the font and the arrows showing the location of Transect I and Transect II thicker to make them standout more.

Figure 2 Make the bar representing the scale (0-500) less prominent. That's one of the first things that draws reader's attention when you look at the map. Instead, highlight A, B, C better, so that the title of each map is not hidden among all the other text on the maps.

Figure 3 Please specify whether the n-alkane data shown in the bar graphs represents all the plants collected along the Transects I and II or only a subset.

Figure 4 Remind the reader what kind of "n-alkane ratio" is plotted here.

Figure 5 Specify that "compound-specific" refers to n-C29 and n-C31 alkanes.

Figure 6 Which homologues were included in the calculation of n-alkanes concentrations? What ratio of n-alkanes are the authors referring to?

---

## Author Comment (AC2) · 17 Dec 2019

**Author responses to the review of Referee #2 of the Biogeosciences manuscript bg-2019-251: 'Leaf wax *n*-alkane patterns and compound-specific δ¹³C of plants and topsoils from semi-arid Mongolia'**

By Julian Struck, Marcel Bliedtner, Paul Strobel, Jens Schumacher, Enkhtuya Bazarradnaa, Roland Zech

We are thankful to referee #2 for the detailed and constructive comments on our manuscript, and we will revise it accordingly.

Please find below our point-to-point response to the review of referee #2. Referee comments are given in *black italic font,* our response to each point is given in blue regular font. Resulting changes are given in *blue italic*.

**Anonymous Referee #2:**

*The authors of this manuscript investigated the link between several environmental parameters (mean annual temperature, mean annual precipitation, and aridity index) and molecular and stable carbon isotope compositions of leaf wax n-alkanes extracted from modern higher plants and topsoils along 2 broad transects in Mongolia. The manuscript provides much needed molecular and stable isotope data for that area and will be of interest to biogeochemists, paleoecologists, and paleoclimatologists studying past climate change in the arid zones of central Eurasia. The manuscript fits within the scope of Biogeosciences Discussions and should be published in this journal provided the authors address the following issues:*

→ We are very happy about this positive comment on our manuscript.

**Major point to address**

*First, lumping topsoil n-alkane data when looking at Transects I and II*

*Transect II The data shown in Fig. 4 and Fig. 5 for Transect II has a lot of scatter. The transect includes 3 different areas A, B, and C, with area B corresponding to an altitudinal transect. Could this scatter be the result of additional factors controlling the molecular and d13C data along the altitudinal transect in addition to those that play role along the W-E transect (i.e. A, through B (only sites 22, 23, 24, 25) through C)? Could the altitudinal transect sites be plotting separately?*

*Transect II + Transect I A similar issue could be the reason for a large scatter in the d13C data in Fig. 6 (top 2 sections). There is a lot of scatter at ∼ - 6C MAT, ∼ 210 mm MAP, and ∼ 0.28 AI. Could this be caused by multiple factors (in addition to those plotted along the X-axis) controlling the d13C values of n-alkanes along and within these transects? Can the data be plotted separately to provide a more nuanced assessment?*

→ Thank you for this comment. The scatter you mentioned is not caused by the samples from the altitude transect (TSII-B), but by the samples from the Telmen catchment (TSII-C) (Fig.1). Fig. 1 shows exemplarily the correlation of δ¹³C *n*-C$_{29}$ with altitude, MAT and MAP, with colours indicating the different sites.

[Figure]

*Figure 1: Compound-specific δ¹³C (n-C₂₉) from Mongolian topsoils correlated against altitude and climatic parameters (MAT, MAP). Red trend lines illustrate linear regressions. Bold values indicate the level of significance (α = 0.05)*

For all leaf wax proxies, we checked for altitude as a controlling factor. There are significant correlations between altitude and $\Sigma$ $n$-alkane (both, $n$-C$_{23}$ – $n$-C$_{35}$ and $n$-C$_{25}$ – $n$-C$_{35}$), OEP and $\delta^{13}$C (both, $n$-C$_{29}$ and $n$-C$_{31}$). Correlation with ACL is weak and non-significant, and the $n$-alkane ratio shows no correlation at all. We will include a detailed description about altitude as a controlling factor within the figures and the discussion part.

Nevertheless, altitude generally controls MAT ($R^2 = 0.624$) and MAP ($R^2 = 0.395$), and we think that all factors are important and influence the leaf wax signal. However, the scatter in our transect is rather the result of site-specific/micro-climatic characteristics and variations in plant physiology. We will strengthen the discussion and possible reasons for the scatter within the manuscript. In case of $\delta^{13}$C ($n$-C$_{29}$), the scatter of TS II – C is caused by the occurrence of succulent plants using the CAM metabolism and are thought to be more enriched in $^{13}$C.

Concerning Fig. 4 and Fig. 5, we see no real added value in separating the altitudinal transect. At this point, we simply separate the dataset in predominantly arid (transect I) and predominantly semi-arid (transect II) to show differences between both environments. However, we understand your argumentation and we will divide the dataset for the scatterplots (Fig. 6) as exemplarily shown here in figure 1.

**Other minor issues**

*Line 34 (here and similar issues throughout the manuscript) "an enrichment of leaf wax d13C" d13C values are numbers. Values can't be enriched or depleted. Please re-phrase. 13C-enriched leaf wax perhaps?*

➔ We will rephrase those sentences or change it to "$^{13}$C enriched".

*Lines 82-83 "the topsoils were sampled together with the dominant plant species, which comprise the woody shrub Caragana spp. …" How the dominance of these species was assessed? Is there any previous study concerning species distribution in the area covered by this project? Or is it a subjective assessment?*

➔ This is a subjective assessment! Different plants were sampled around the soil sampling sites (~5 m²). Those plants were sampled individually and determined by a botanist at the Institute of Plant and Agricultural Sciences, Mongolian University of Life Sciences, Darchan, Mongolia.

*Line 90 "Total lipids … of plants" What part(s) of plant was(were) extracted? Just leaves or was it together with the stem and roots?*

➜ For lipid extraction, just the leaves/needles were used, except for the grasses where we used the entire grass without roots.

*Line 107 "were 0.1 per mil the standard deviation" Is there an "and" missing? Also, does it make sense to report d13C values in Suppl. Mat to the second digit after the decimal point, if the reported std. dev. is 0.1 per mil, i.e. no better than the first digit after the decimal point?*

➜ This was truly a mistake! The std. dev. for the topsoils and plants was better than 0.66 per mil (0.7) for both compounds. We will change the sentence as follows:

➜ *The  standard deviation for the triplicate measurements were < 0.7‰ and the standard deviation for the alkane standards was better than 0.2‰ (n = 102).*

*Lines 110-111 "n-alkane concentrations … were calculated as the sum of n-C25 and n-C35" Why was n-C23 excluded? It is a major n-alkane in Larix sp.*

➜ The sum of $n\text{-}C_{25}$ to $n\text{-}C_{35}$ typically comprises the $n$-alkanes within the leaf waxes of higher terrestrial plants. It is true that *Larix* has their dominance in $n\text{-}C_{25}$ and $n\text{-}C_{23}$. However, differences between the concentration calculated from $n\text{-}C_{25}$ - $n\text{-}C_{35}$ and $n\text{-}C_{23}$ - $n\text{-}C_{25}$ are minor and $n\text{-}C_{23}$ and $n\text{-}C_{25}$ are not dominant in the respective topsoils, but we will implement this data within the supplements.

*Lines 118-119 "A normalised n-alkane ratio … n-C29 and n-C31"; lines 177-178 Please explain the significance of this ratio. If this refers to trees/shrubs vs. grasses, why not to include n-C27 and n-C33, respectively?*

➜ We have chosen this normalized n-alkane ratio, because $n\text{-}C_{29}$ and $n\text{-}C_{31}$ are the most dominant $n$-alkanes for the grasses as well as for the shrubs *Caragana spp.* and *Artemisia spp.* For those plants, $n\text{-}C_{27}$ and $n\text{-}C_{33}$ are not the dominant chain-lengths and allow no separation in terms of grasses vs. shrubs, which you can see in the ACL that include $n\text{-}C_{27}$ and $n\text{-}C_{33}$.

*Line 165 "in line with previous regional studies" Specify what regions were covered by these studies previously? Is it similar to the region covered in this project?*

➜ Cheung et al. (2015) and Wang et al. (2018c) are located on the Tibetan Plateau, Liu et al. (2018) on the Chinese Loess Plateau and Bliedtner et al. (2018) in the Caucasus region. Concerning the environmental conditions, we think that the studies of Wang et al. (2018c) and Liu et al. (2018) are comparable to our study. Cheung et al. (2015) and Bliedtner et al. (2018) receives with 480 mm/a and up to 1800 mm/a higher amounts of precipitation compared to Mongolia. We will specify the regions of the previous studies in the manuscript.

*Line 169 "the findings of Wang et al. (2018b) from China" What part of China? It is a big country with multiple climatic and ecological zones.*

➜ The investigated transect of Wang et al. (2018b) covers the 400 mm isohyet in China. More specifically, it follows Inner Mongolia towards the Tibetian Plateau. We will change "from China" to "along a transect from north-western to central China".

*Line 202-233 Section "The leaf wax signal from plants to topsoils along transect II" This section could be broken down into several paragraphs to make it easier to follow. Also, why not to give a number for this and the next (starting on line 234) subsections? 4.3.1 and 4.3.2 perhaps?*

➜ We agree with this point and will implement this suggestion.

*Lines 223-224 "Compared to the plants, … d13C isotopes of the topsoils are slightly more enriched" What does "d13C isotopes" mean? Please rephrase.*

➔ The isotopic $\delta^{13}C$ signature of the topsoils from transect II. Will be specified.

*Also, I don't see this from the graphs in Fig. 5. The d13C values of n-C29 alkane in soils aren't really that different from those in Caragana and Larix spp. The absence of this "enrichment" is particularly evident when looking at the d13C values of n-C31.*

➔ Concerning the topsoils of transect II, differences between the topsoils (transect II) and plants reveal no statistical significance. However, the median of both compounds ($n$-$C_{29}$ and $n$-$C_{31}$) is slightly enriched compared to the plants: For $n$-$C_{29}$ it's up to 1.7‰ for $n$-$C_{31}$ it's up to 1.5‰.

*Line 237 "chapter 4.4" I'd call it a section rather than a "chapter".*

➔ We agree!

*Lines 248-249 "are in agreement with climatic control and the fact that higher temperatures reduce the decarboxylation pathway and the formation of n-alkanes (Shepherd and Griffiths 2006)" I don't think this explanation works here. The cited paper evaluates the effect of stress on various factors that control leaf wax biosynthesis within plants. The subject matter of this section is n-alkane content of soils. There are multiple other reasons, in addition to the biochemical ones within the plant, that could play a role in the distribution of n-alkanes along the transects studied.*

➔ You are right, there are many other reasons playing a role, like *n*-alkane degradation, biomass productivity or livestock grazing, which we have discussed in this paragraph. Thus, we will delete this hypothesis to reduce confusion.

*Lines 257-258 "In contrast, compound-specific d13C correlated significantly with climatic parameters." There is a lot of noise in the d13C data. Please discuss possible reasons for the scatter. See the MAJOR POINT above.*

➔ Is discussed above.

*Line 272 "Mongolian plants show" It is a peculiar way of referring to these plants. Using their species names and mentioning that they were sampled in Mongolia would be a better way of describing them.*

➔ We agree and we will adapt this!

*Lines 276-277 "for reconstructing vegetation changes in Larix sp." Sounds awkward. Please re-phrase.*

➔ We will rephrase it as follows:
➔ *'However, Larix sp. produces only few amounts on n-alkanes and their dominance of mid-chain n-alkanes are not distinct in the respective topsoils. Thus, n-alkanes are not useful for reconstructing changes in the abundance of Larix sp.'.*

**FIGURES**

*Figure 1 Please remind the reader what SRTM DEM stands for so that there is no need to look for this information in the text. Also, make the font and the arrows showing the location of Transect I and Transect II thicker to make them standout more.*

➜ We agree with this point and add an explanation within the figure caption.
➜ Fonts and arrow thickness will be adapted

*Figure 2 Make the bar representing the scale (0-500) less prominent. That's one of the first things that draws reader's attention when you look at the map. Instead, highlight A, B, C better, so that the title of each map is not hidden among all the other text on the maps.*

➜ We agree with this point and will change it accordingly.

*Figure 3 Please specify whether the n-alkane data shown in the bar graphs represents all the plants collected along the Transects I and II or only a subset. Figure 4 Remind the reader what kind of "n-alkane ratio" is plotted here.*

➜ All the plants were sampled along transect II. We will highlight this information within the figures caption.
➜ We will add the equation of the n-alkane ratio within the figure.

*Figure 5 Specify that "compound-specific" refers to n-C29 and n-C31 alkanes.*

➜ We will adapt this.

*Figure 6 Which homologues were included in the calculation of n-alkanes concentrations? What ratio of n-alkanes are the authors referring to?*

➜ Both information will be added to the figure or at least within the caption.